# MALS-Net: A Multi-Head Attention-Based LSTM Sequence-to-Sequence Network for Socio-Temporal Interaction Modelling and Trajectory Prediction

**DOI:** 10.3390/s23010530

**Published:** 2023-01-03

**Authors:** Fuad Hasan, Hailong Huang

**Affiliations:** Department of Aeronautical and Aviation Engineering, The Hong Kong Polytechnic University, Hong Kong, China

**Keywords:** vehicle trajectory prediction, autonomous driving, LSTM, transformer, multi-head attention

## Abstract

Predicting the trajectories of surrounding vehicles is an essential task in autonomous driving, especially in a highway setting, where minor deviations in motion can cause serious road accidents. The future trajectory prediction is often not only based on historical trajectories but also on a representation of the interaction between neighbouring vehicles. Current state-of-the-art methods have extensively utilized RNNs, CNNs and GNNs to model this interaction and predict future trajectories, relying on a very popular dataset known as NGSIM, which, however, has been criticized for being noisy and prone to overfitting issues. Moreover, transformers, which gained popularity from their benchmark performance in various NLP tasks, have hardly been explored in this problem, presumably due to the accumulative errors in their autoregressive decoding nature of time-series forecasting. Therefore, we propose MALS-Net, a Multi-Head Attention-based LSTM Sequence-to-Sequence model that makes use of the transformer’s mechanism without suffering from accumulative errors by utilizing an attention-based LSTM encoder-decoder architecture. The proposed model was then evaluated in BLVD, a more practical dataset without the overfitting issue of NGSIM. Compared to other relevant approaches, our model exhibits state-of-the-art performance for both short and long-term prediction.

## 1. Introduction

Autonomous driving has been gaining an increasing interest due to its guarantee of safer driving on the road. Recently, one of the core functions of autonomous driving that has been of particular interest in the literature is the future trajectory prediction of neighboring vehicles. In a congested highway driving scenario, where all cars are often driving at very high speeds, even a mildly reckless maneuver can consequentially result in a ripple effect causing a serious accident, which is why an autonomous car needs to forecast the future trajectories of its surrounding vehicles to assess the risk of its own future maneuvers.

However, predicting the future trajectories of surrounding vehicles has been extremely challenging in practice since the prediction is not only dependent on the historical trajectories of the vehicles but also the dynamic and complicated socio-temporal interdependence [1] of the dense web of vehicular traffic around the car, including other cars, buses, trucks, etc. For instance, in a dense highway setting, if one driver tries to change the lane, the neighboring lane driver might slow down to give way and the current lane driver will speed up to take the driver’s place. Hence, to make an accurate prediction of the future trajectory, a proper model of the interaction between the participants needs to be utilized as one of the parameters for the prediction besides only raw historical trajectories of the surrounding vehicles.

To tackle this problem of complexity in modeling interaction, recent few years have seen the emergence of a plethora of deep learning-based data-driven models. To address the challenge of modeling the influence of neighboring vehicles on the target or ego vehicle, i.e., the social interaction, typical deep-learning-based operations that have been explored include various pooling techniques [2] with convolution [3] or graphs or both [4]. These have often been coupled with an RNN such as LSTM [5] or GRU [6] to exploit its inherent memory capability and extract temporal correlation. However, these models often struggle to model complex long-term temporal correlations. Transformers have been introduced to the literature with the promise of tackling the issue of long-term temporal correlation as well as parallelizing the decoding process. Inspired by its distinct attention mechanism, various attention-based techniques have been adopted in [1,7]. However, the multi-headed attention mechanism, originally proposed in the traditional transformer [8], has not extensively been explored in the highway trajectory prediction problem, mainly due to the problem of accumulative errors resulting from the autoregressive decoding procedure of transformers [9].

Moreover, most studies related to highway future trajectory prediction have mainly adopted the popular NGSIM dataset [10] to evaluate their performance. The NGSIM dataset was mainly collected to satisfy the need for data with explicit microscopic traffic flow and car-following information, for advancing traffic flow theory [11]. However, the NGSIM dataset was discovered to be extremely noisy in the literature by [12,13,14]. Various efforts have been made to smooth out the noise such as low pass filter, interpolation, etc. by [15]. Ref. [11] demonstrated that despite these efforts, the problem still persists and training on the NGSIM dataset thus has the potential possibility of resulting in a model with overfitted parameters, making it unfeasible to be deployed for practical application.

In this study, we thus propose a transformer-based LSTM sequence-to-sequence model to tackle the interaction-aware trajectory prediction problem. We exploit the main mechanism of the transformer, the multi-head attention (*MHA*), and implement it on an *MHA*-LSTM fused sequence-to-sequence architecture to tackle the autoregressive accumulative error problem of the transformer decoder. The encoder in our model is capable of encoding both the social and temporal interaction of the vehicles by implementing two Multi-Head Attention layers in the encoding process to put attention on both vehicle-vehicle and timestep-timestep graphs, followed by a traditional LSTM layer. This is then followed by another Multi-Head Attention-based decoder with an LSTM layer that is be able to decode by focusing on the socio-temporal interaction between the vehicles in successive decoding steps, which helps prevent the model from accumulating autoregressive decoding errors and also improve future trajectory prediction accuracy. We thus summarize the contribution and the academic as well as the practical novelty of our original work as follows.

In order to model the social dependence of past trajectories, we propose a Social Multi-Head Attention (*SMHA*) mechanism, and to model the temporal dependence, we use a successive Temporal Multi-Head Attention (*TMHA*) mechanism to focus attention on both social and temporal interaction and encode the input data.A similar *MHA*-based LSTM decoding step is proposed to extract the predicted socio-temporal interaction in successive decoding steps, which improves the successive prediction accuracy and minimizes the accumulative errors of the transformer decoder.The evaluation of the method has been performed on BLVD, a large-scale vehicle trajectory dataset that has less noise and is extracted from egocentric onboard-sensor data, to prevent overfitting. The experimental results demonstrate the superior performance of our model over state-of-the-art methods that have been implemented on both NGSIM and BLVD datasets.

The remainder of this article is constructed as follows. Section 2 provides an overview of the literature review on the general vehicle future trajectory prediction problem. Section 3 formulates the main trajectory prediction problem that we focus on and also provides an in-depth look on the model we propose. Section 4 mentions the implementation details and also reports our experimental results from both comparative and ablative studies. Finally, Section 5 concludes this paper and discusses future directions.

## 2. Related Work

The very first versions of the trajectory prediction literature focused on multiple physical model-based techniques. These essentially predict the future motion of a vehicle based on a form of kinematics, statistics or a fusion of both models. Some popular approaches include the Constant Velocity (CV) [16] and Constant Acceleration model (CA) [17], Kalman Filter-based motion model [18], Gaussian Mixture Model (GMM) [19] and Hidden Markov Model (HMM) [20]. This allowed [21] to utilize CV and CA to develop an intelligent driver model. Further works by [22] proposed a fusion of GMM with HMM to use parameters such as vehicle motion estimation, traffic patterns, and spatial interaction to predict the motion of the surrounding vehicles. The purpose of predicting the trajectories of surrounding or sometimes the ego vehicle has mainly been done to carry out risk assessments for safe automated driving, such as in [23,24], where Rapid Random Trees (RRT) and Linear Matrix Inequality (LMI) has often been implemented to analyze driving risk of lane change maneuvers or construct a human-machine shared control system [25]. However, it has been observed that physical models have a very short prediction horizon, beyond which the accuracy of the model becomes unfeasible [26]. Therefore, this has pushed the research toward the direction of data-driven methods.

The early works on data-driven models explored simpler learning-based models for interaction-aware trajectory prediction, such as support vector machines in [27,28] and Gaussian process regression [29]. However, these models need a handful of manually designed features to be completely constructed. As the incresing complexity of Spatio-temporal interdependence modelling was recognized, it became unfeasible to use handcrafted features and hyperparameters for these models. Thus deeper learning models, which can design and train their own features, proved to be a breakthrough.

At first, most deep learning models mainly focused on extracting temporal correlation via RNN architectures such as LSTM [30] and GRU [31], only focusing on the historical trajectory to make the future prediction [32,33]. Other deep-learning approaches adopted LSTMs to utilize their time-series modelling ability to make an energy-aware driving behavior analysis as well as predict the motion [34]. Due to the encoder-decoder sequence-to-sequence architecture of RNNs, they lacked the ability to model any sort of social or spatial interaction. More recent literature has thus mostly focused explicitly on coupling RNNs with some sort of social feature extraction architecture to model vehicle interaction. An approach adopted by [35] proposed a statistical method GMM fused with LSTMs to generate the leading-vehicle trajectory. Most popular such approaches, however, have mostly focused on other neural architectures such as CNNs and GNNs.

One of the widely used architectures to model social interaction has been the convolution and social pooling approach. One of the first social pooling approaches was proposed in [2] which used an LSTM along with a social pooling strategy (S-LSTM) to decode the prediction. In [36], a social GAN (SGAN) was used which utilizes both sequence-to-sequence architecture and a GAN to make the final prediction. A convolutional social pooling approach was proposed in [3] where the convolutional kernel-striding was applied to social pooling to improve on the LSTM social pooling approach in [2]. A CNN-LSTM approach was proposed in [37] which used an LSTM encoder-decoder architecture and performed a CNN operation on the hidden-layer tensors.

Graph-based architectures have also been frequently used for predicting future trajectories. Modelling the interaction-aware trajectory prediction of surrounding vehicles as graphs with nodes being the vehicles and the edges being the interaction between them has often been implemented in the literature. Such an approach was implemented in [6] which constructed a GNN to make the future trajectory prediction.

Attention-based networks, however, have been a recent breakthrough, because of their ability to rapidly extract important information from historical tracks. Ref. [38] applies a message-passing architecture to focus on pedestrian motion in order to make a future prediction. Another model which also focuses on pedestrian trajectory prediction is [39], wherein two attention layers are stacked to extract both spatial and temporal attention. Graphs have also been utilized in the attention mechanism. The reference [40] utilizes social graph attention to model both social and temporal interaction based on relative positions. However, the multi-head attention mechanism of the transformer does not appear to have been used as extensively in vehicle trajectory prediction. Its utilization mainly spans pedestrian trajectory prediction.

## 3. Methodology

### 3.1. Problem Formulation

In this paper, the future vehicle trajectory prediction problem is formulated as a non-linear regression task where the inputs to the model are the past trajectories of the observed neighbouring vehicles, which can be represented by
(1)$$X=\left\{p_{1},p_{2},\ldots ,p_{T}\right\},$$
where
(2)$$p_{t}=\left\{\left(x_{t}^{0},y_{t}^{0}\right),\left(x_{t}^{1},y_{t}^{1}\right),\ldots ,\left(x_{t}^{N},y_{t}^{N}\right)\right\}.$$

$p_{t}^{i}=\left(x_{t}^{i},y_{t}^{i}\right)$ are the coordinates *x* and *y* of vehicle *i* at timestep *t*, where t∈[1,T] and i∈[1,N], *T* stands for the total length of the observed trajectory and *N* is the total number of observed neighbouring vehicles within 30 meters of the target vehicle, i.e., $\left|p_{t}^{target}-p_{t}^{i}\right|\leq 30$.

Based on the past trajectories, the objective of the proposed model is to learn a non-linear regression function that predicts the coordinates of all observed vehicles in the prediction horizon *F*, such that the predicted trajectories of the neighbouring vehicles are represented as $\hat{Y}$, as follows, which approximates the ground truth (GT) trajectory, $Y=\left\{q_{T+1},q_{T+2},\ldots ,q_{T+F}\right\}$
(3)$$\hat{Y}=\left\{{\hat{q}}_{T+1},{\hat{q}}_{T+2},\ldots ,{\hat{q}}_{T+F}\right\},$$
where
(4)$${\hat{q}}_{t}=\left\{\left({\hat{x}}_{t}^{0},{\hat{y}}_{t}^{0}\right),\left({\hat{x}}_{t}^{1},{\hat{y}}_{t}^{1}\right),\ldots ,\left({\hat{x}}_{t}^{N},{\hat{y}}_{t}^{N}\right)\right\}.$$

The frame of reference of the dataset used in this paper is by default ego-centric as the trajectories were extracted from the onboard sensors. The longitudinal y-axis indicates the direction forward to the ego vehicle and the lateral x-axis direction indicates the axis perpendicular to it. The right-hand side is considered positive according to the dataset [41]. As a result, the model is independent of the curvature of the road which conveniently allows it to be used in the highway as long as an object-detection and a lane estimation algorithm is built on the target vehicle [3].

### 3.2. Network Architecture

The high-level architecture of the model we propose is shown in Figure 1. As discussed previously, it is structured as a sequence-to-sequence architecture capable of extracting socio-temporal correlation from past observed trajectories and then generating future trajectories based on that. Raw input $X=[p_{1},p_{2},\ldots ,p_{T}]$ is first preprocessed into a suitable tensor. The encoder module then encodes the socio-temporal attention data into the input via two distinct multi-head-attention layers and then also encodes temporal memory via the LSTM layer. The decoder then decodes the encoded information into future predictions $\hat{Y}=\left[{\hat{q}}_{T+1},{\hat{q}}_{T+2},\ldots ,{\hat{q}}_{T+F}\right]$. The hidden features are passed onto successive decoding steps via further *MHA* layers to improve further future predictions which also prevents the traditional transformer decoding accumulation error by enriching the hidden features.

### 3.3. Input Representation

To input the trajectory data into our proposed model, the coordinates are first scaled to a range of (−1, 1) to aid the model to reach faster convergence [42]. The normalized coordinates are then preprocessed into a tensor $X\in R^{N\times T\times D}$ where D is the feature space, in the case of our input having a value of 2 representing the (x,y) coordinates of the vehicles.

This tensor is then run through two different embedding layers simultaneously. The first embedding layer *PE* (see Figure 1 above), is a multi-layer perceptron (MLP) which, as shown in Equation (Equation 5), maps raw input features to a higher dimension $D^{IE}$. The second embedding layer *DE*, in Equation (Equation 6), maps the relative distance among each vehicle $p_{t}^{i}-p_{t}^{j}$, for i,j∈[1,N] and t∈[1,T], to a higher dimension $D^{DE}$. The two feature representations are then concatenated to produce the final input into the model with a feature space size $D_{model}=D^{IE}+D^{DE}$. The final input thus consists of both the embedded position and relative distance between vehicles as features before being run through the model.
(5)$${IE}_{t}^{i}=\mathrm{MLP}\left(p_{t}^{i},W_{IE}\right)$$
(6)$${DE}_{t}^{i}=\mathrm{MLP}\left(p_{t}^{i}-p_{t}^{j},W_{DE}\right)$$

### 3.4. Social Multi-Head Attention

The Social Multi-Head Attention (*SMHA*) layer, as shown in Figure 2, is proposed to extract the rich inter-vehicular social interaction information from the input. The features of the embedded input $X_{emb}\in R^{N\times T\times D_{model}}$, through the last *IE* and *DE* layers, now contain rich embedded higher dimensional information about both the trajectories and relative distance of each vehicle. However, there is a multitude of features and extracting a viable correlation requires putting greater weight on features that actually contribute to a unit change in results. This work is done by an attention layer. This first multi-head attention layer enables the model to understand what features to put the most attention to. The embedded input is first transposed to ${\left(X_{emb}\right)}^{T}\in R^{T\times N\times D_{model}}$ so that we can extract attention weights from the last two dimensions $R^{N\times D}$. The features of the input are then split into $n_{heads}$ transformer heads as $R^{n_{heads}\times T\times N\times \frac{D_{model}}{n_{heads}}}$ and fed into the Multi-Head Attention (*MHA*) layer. The *MHA* first computes the query (Q), key (K), and value (V) matrices at timestep *t*, as shown in Equation (Equation 7), via three MLPs which conserves the size of the feature space of the input as $\frac{D_{model}}{n_{heads}}$.
(7)$$Q_{t}=\mathrm{MLP}\left({\left\{x_{t}^{i}\right\}}_{i=1}^{N},w_{q}\right),\;K_{t}=\mathrm{MLP}\left({\left\{x_{t}^{i}\right\}}_{i=1}^{N},w_{k}\right),\;V_{t}=\mathrm{MLP}\left({\left\{x_{t}^{i}\right\}}_{i=1}^{N},w_{v}\right)$$

The weights $w_{q},w_{k}$ and $w_{v}$ are initialized as samples from the Xavier Normal Distribution [43], $N(0,\sigma^{2})$ where
(8)$$\sigma =k\times \sqrt{\frac{2}{l_{in}+l_{out}}}.$$

The value of *k* is the gain designed to be 1 and $l_{in}$ and lout are the input and output feature dimensions which in this case are both $\frac{D_{model}}{n_{heads}}$.

The inter-vehicular interaction is then represented by an undirected graph $G_{t}=\left\{V_{t},E_{t}\right\}$ where the nodes $V_{t}$ represent the observed vehicles $V_{t}=\left\{v|i=1,2,\ldots ,N\right\}$ and the edges represent the interaction as binary values between vehicles *i* and *j*, $E_{t}\left[i\right]\left[j\right]=1$ if $p_{t}^{i}-p_{t}^{j}\leq d_{near}$, otherwise the value is zero. The interaction graph is used to mask social attention.

The masked attention of head *h* at timestep *t* is then computed as
(9)$${\mathrm{Attention}}_{h}\left(Q_{t},K_{t},V_{t}\right)=\frac{\mathrm{Softmax}\left({\left[{\left(q_{t}^{i}\right)}^{T}k_{t}^{j}\right]}_{mask}\right)}{\sqrt{d_{k}}}{\left[v_{t}^{i}\right]}_{{mask}^{\prime }}$$
where $mask=\left\{j|G_{t}\left[i,j\right]=1,j\in \left[1,N\right]\right\}$. The masking based on the interaction graph $G_{t}$ allows the attention scores to be calculated only when the vehicles are near the target, defined by the threshold $d_{near}$. In practicality, this is done to resemble a real driving scenario. A driver’s decision is only based on cars that are observable and nearby. If it cannot observe a car and/or it is not nearby, it is not practical to calculate an interaction score between them as they do not share an interaction in that scenario. Therefore, we have to design the value of threshold $d_{near}$ to resemble a region around the driver where the driving decision would be affected. The masked Multi-Head Attention on the social correlation can then be computed as
(10)$$\mathrm{MHA}\left(Q_{t},K_{t},V_{t}\right)=\mathrm{Concat}\left(Attention_{1},\ldots ,Attention_{n_{heads}}\right).$$

The output from the Multi-Head Attention layer of size $O_{MHA}\in R^{N\times T\times D_{model}}$ is then fed through an intra-layer MLP which helps boost training speed. This is then followed by an Add & Norm layer. The Add layer is simply a residual connection of the transposed input ${\left(X_{emb}\right)}^{T}\in R^{T\times N\times D_{model}}$ which adds the current output with the input. This connection greatly helps models such as transformers reach convergence faster by resolving the vanishing gradient problem so it is also extensively used in our model [44]. The output of the Social Multi-Head Attention Layer is thus computed as follows,
(11)$$O_{SMHA}=\mathrm{Norm}\left(\mathrm{MLP}\left(O_{MHA}\right)+{\left(X_{emb}\right)}^{T}\right).$$

The output $O_{SMHA}$ has the same dimensions as the transposed Input ${\left(X_{emb}\right)}^{T}\in T\times N\times D_{model}$.

### 3.5. Positional Encoding

To date, the *SMHA* layer has extracted the social correlation via the weights and biases of the layer. Thus, we now have to extract the temporal correlation from $O_{SMHA}$. However, in order to model temporal dependencies, the order is very important. Even though the multi-head attention mechanism is powerful enough to process a long sequence of data very quickly due to its attention graph-based architecture, in doing so, it loses its sense of the order of each point in a sequence. The order was not particularly useful in the *SMHA* step as it is redundant information for social interaction between vehicles, but it is, however, imperative for the output $O_{SMHA}\in R^{T\times N\times D_{model}}$ to preserve its sense of order before being pushed through the *TMHA* step, otherwise it will produce huge errors in the decoding step. Therefore, we propose a Positional Encoding layer (*PE*) to inject the sequential order into the data via sinusoidal positional encoding. First we transpose $O_{SMHA}$ back to $O_{SMHA}\in R^{N\times T\times D_{model}}$ and then initialize the positional encoding matrix $P\in R^{T\times D_{model}}$. Each scalar $P_{t}^{d}|t\in \left[0,T\right],d\in \left[0,D_{model}\right]$ of the Positional Encoding Matrix is a sinusoidal function as follows
(12)$$\left\{\begin{array}{c} {\mathrm{PE}}^{i}\left(t,2d\right)=\sin\left(\frac{t}{{10,000}^{\frac{2d}{D_{model}}}}\right),\\ {\mathrm{PE}}^{i}\left(t,2d\right)=\cos\left(\frac{t}{{10,000}^{\frac{2d}{D_{model}}}}\right). \end{array}\right.$$

The above operation is then applied for t∈[1,T] and then the positional encoding matrix $P\in R^{T\times D_{model}}$ is added to each vehicle i∈[1,N], to obtain $O_{PE}\in R^{N\times T\times D_{model}}$, which is the sequential order infused input to the *TMHA* layer.

### 3.6. Temporal Multi-Head Attention

In order to model and extract the temporal correlation of each vehicle from the input trajectory data, we propose another multi-head attention layer (*TMHA*), as illustrated in Figure 3, similar to the *SMHA*. The feature space of $O_{PE}$ is first split into $n_{heads}$ transformer heads as $R^{n_{heads}\times T\times N\times \frac{D_{model}}{n_{heads}}}$ and fed into a Multi-Head Attention Layer (*MHA*). The *MHA* first feeds the input into three MLPs respectively in order to compute the query (Q), key (k), and value (V) matrices for vehicle *i*, as shown in Equation (Equation 13), which preserves the feature space of the input as $\frac{D_{model}}{n_{heads}}$.
(13)$$Q_{i}=\mathrm{MLP}\left({\left\{o_{t}^{i}\right\}}_{t=1}^{T},w_{q}\right),\;K_{t}=\mathrm{MLP}\left({\left\{o_{t}^{i}\right\}}_{t=1}^{T},w_{k}\right),\;V_{t}=\mathrm{MLP}\left({\left\{o_{t}^{i}\right\}}_{t=1}^{T},w_{v}\right)$$

The weights $w_{q},w_{k}$ and $w_{v}$ are again initialized as samples from the Xavier Normal Distribution [43] again, $N(0,\sigma^{2})$, as discussed in Equation (Equation 8) above.

The masked attention of head *h* for the *i*-th vehicle is then computed as:(14)$${\mathrm{Attention}}_{h}\left(Q_{i},K_{i},V_{i}\right)=\frac{\mathrm{Softmax}\left({\left[{\left(q_{t}^{i}\right)}^{T}k_{u}^{i}\right]}_{u=1}^{T-1}\right)}{\sqrt{d_{k}}}{\left[v_{t}^{i}\right]}_{u=1}^{T-1},$$
where $\left({\left[{\left(q_{t}^{i}\right)}^{T}k_{u}^{i}\right]}_{u=1}^{T-1}\right)$ demonstrates the time-masking of the timesteps. Essentially, the time mask prevents the current steps from accessing features from the relative future. For example, if the current timestep is *s* then the features from timestep *s* to *T* will be masked as zero. This prevents the model from overfitting and making exclusive correlations on the training data.

The Multi-Head Attention to the temporal dependency can then be calculated as
(15)$$\mathrm{MHA}\left(Q_{i},K_{i},V_{i}\right)=\mathrm{Concat}\left(Attention_{1},\ldots ,Attention_{n_{heads}}\right).$$

The output from the Multi-Head Attention layer of size $O_{MHA}\in R^{N\times T\times D_{model}}$ is then fed through an intra-layer MLP, followed by an Add & Norm layer, which provides the residual connection of the output from the last layer $O_{PE}\in R^{N\times T\times D_{model}}$. After the residual connection and normalization, the output is fed through another multi-layer perceptron network (*FFN*). *FFN* is made up of two feed-forward networks that successively map the feature vectors to a higher dimension and then back to the original dimension such that output from MLP1 is $R^{N\times T\times D_{FFN}}$ and MLP2 maps it back to $R^{N\times T\times D_{model}}$. This is done to add more model parameters so that the temporal attention vectors can be fed through further layers such as an RNN. The output of the Temporal Multi-Head Attention Layer is thus computed as follows,
(16)$$O_{TMHA}=\mathrm{FFN}\left(\mathrm{Norm}\left(\mathrm{MLP}\left(O_{MHA}\right)+O_{PE}\right),W_{FFN}\right),$$
where
(17)$$\mathrm{FFN}=MLP2(MLP1\left(FFN_{in},w_{MLP1}\right),w_{MLP2}).$$

Thus the output from the *TMHA* layer now has both social and temporal correlation encoded into it. We refer to the *SMHA*, *PE* and *TMHA* layers compounded together as *STMHA*. We then stack *M* number of *STMHA* layers successively to extract more complex socio-temporal dependence from the past trajectory information.

### 3.7. LSTM Encoder

We propose a traditional LSTM encoder after the *STMHA* layer stack to extract the hidden memory information from the output tensor $O_{STMHA}\in R^{N\times T\times D_{model}}$. This is done to facilitate the hidden tensor passing to the LSTM decoder module. We propose this sequence-to-sequence architecture as a replacement for the traditional transformer architecture with an *MHA*-based transformer decoder. As mentioned before, due to the autoregressive nature of the transformer decoder, it carries the problem of accumulated errors. We thus use a sequence-to-sequence LSTM decoder to solve this problem. At first, the LSTM encoder layer recurrently computes the hidden-state tensor $H_{t}\in R^{L\times N\times D_{hid}}$ at time *t* for t∈[1,T] with *L* encoder layers and $D_{hid}$ hidden dimensions as
(18)$$o_{t}^{i},h_{t}^{i}=\mathrm{LSTM}\left(h_{t-1}^{i},o_{t-1}^{i},w_{t-1}^{i}\right).$$

After recurrently updating the hidden state for *T* timesteps, the $H_{T}\in R^{L\times N\times D_{hid}}$ at the final timestep *T* is then passed on to the decoder module.

### 3.8. STMHA-LSTM Decoder

To decode the hidden-state tensor $H_{T}\in R^{L\times N\times D_{hid}}$ and at the same time extract the rich socio-temporal dependence information encoded into the hidden tensor from the decoded information, we propose an *STMHA*-LSTM decoder, as demonstrated in Figure 4. This module is made up of an LSTM decoder architecture with a built-in multi-head attention mechanism to focus on the predicted behavior to improve further predictions at successive timesteps. The first layer of this architecture is another ordinary LSTM decoder which takes an input and a hidden tensor and decodes the output and updates the hidden tensor. The hidden input that is initialized for the decoder is the hidden tensor from the encoder, $H_{T}$, that was recurrently updated for timesteps [1,T]. As input, the decoder takes in the raw input data at timestep *T*, $X_{T}\in R^{N\times 1\times 2}$ with the last dimension denoting the coordinates of the position of the observed vehicles. It then decodes $H_{T+1}\in R^{L\times N\times D_{hid}}$ for the next timestep T+1, as follows,
(19)$${\hat{q}}_{T+1}^{i},H_{T+1}^{i}=\mathrm{LSTM}\left(H_{T}^{i},o_{T}^{i},w_{T}^{i}\right).$$

The hidden tensor at timestep T+1 is then fed into an *STMHA* layer to extract and update the socio-temporal dependence at the next decoding step. This was repeated for every following decoding step. This is done to further improve the accuracy of successive decoding steps, as well as decode the future trajectories from the hidden information of the transformer encoder with lower auto-regressive accumulated errors. Teacher-forcing is also applied to improve convergence. Each decoding step either takes the predicted output as inputs such as ${\hat{q}}_{T+1}$ or $X_{T+1}$ depending on the teacher forcing ratio. This enables the model to decode the highly accurate future coordinates of each vehicle at every timestep.

## 4. Experiments

In this section, we will report our chosen parameters, hardware, and results from the experiments performed on the publicly available dataset BLVD in order to evaluate the performance of the predictions by our proposed MALS-Net architecture. We will also conduct comparative experiments with other state-of-the-art models as well as ablative comparisons with our chosen architecture.

### 4.1. Dataset

The proposed model was trained, validated, and tested on the BLVD dataset. It consists of a total of 654 high-resolution videos resulting in 120,000 points of data. It was extracted from Changshu city, Jiangsu province. Each frame consists of the ID, 3D coordinates, vehicle direction, and interactive behavior information of all observed vehicles by the ego. Due to the data being collected by onboard sensors, there is less filter noise in the data compared to NGSIM, with more realistic sensor noise, therefore making it more practical. To divide the dataset into training, validation, and testing sets, we follow [41]. The dataset contains various scenario categories of the ego vehicle. We only choose the scenario involving highways with both a high and low density of participants. Other than that, the dataset is also split between day and night. We concatenate and shuffle all these sub-datasets before feeding them into our model.

### 4.2. Implementation Details

We extensively used a desktop running Windows 11 with 3.8 GHz AMD Ryzen 7 CPU, 32 GB RAM, and an NVIDIA 3070Ti Graphics Card to build our model. To train our model, we utilized the parallel High-Performance Computing Service which is a Hong Kong Polytechnic University resource. The high-performance computing nodes are made up of several industry-grade GPUs, the exact model of which is unknown to the authors.

### 4.3. Hyperparameter Settings

For the observable region, we set it to be a 30 m radial region, based on the assumption that a human driver would not be able to see beyond a 30 m region around them. For the interaction graph $G_{t}$, we set the threshold $d_{near}$ to be 15 m. We set the number of *STMHA* layers M=2 and $n_{heads}$ of all the *MHA* modules to be 4 and the value of $D_{model}$ in our model to be 128. The LSTM blocks all have a $D_{hid}$ of 60. and a number of layers, *L* = 2. For the trajectory, we used the past timesteps *T* to be 3 s and future timestamps F to be 5 s. For model training, we used the Adam [45] optimizer with η = 0.001, β = 0.999. The learning rate used is 0.0001 and the batch size of 32. The teacher-forcing ratio used is 0.5.

### 4.4. Evaluation Metrics

In line with the existing literature [1,2,3,4,5,6,36,37,39,40] we adopted the root mean squared error (RMSE) between the prediction and the ground truth for ease of comparison of the model’s performance with other state-of-the-art methods on the NGSIM dataset as well as for analyzing its performance with ground truth. RMSE at prediction time $t^{\prime },t\in \left[T+1,\ldots ,T+F\right]$ can be calculated as follows,
(20)$${\mathrm{RMSE}}_{t}=\sqrt{\frac{1}{L}\sum_{l=1}^{L}{\left({\hat{Y}}_{t}^{l}-Y_{t}^{l}\right)}^{2}},$$
where $\hat{Y}$ is the predicted positions and *Y* is the Ground Truth Position of the *l*-th testing sample at timestep $t^{\prime }$ and *L* is the total length of the test set.

### 4.5. Ablative Analysis

To defend the effectiveness of our architecture, we perform a variety of different ablative experiments in this section. At first to verify the performance of our encoder module in the first encoding social correlation, then the temporal correlation and then memory information into the input, we carry out three distinct experiments. These three experiments are described as follows.

**MALSwoTA:** This variant of the model excludes the *TMHA* layer which extracts the temporal correlation. It thus also excludes the *PE* module and the output from the *SMHA* is directly fed into the LSTM encoder.**MALSwoSA:** This variant of the model excludes the *SMHA* layer which extracts the social correlation. The *DE* module is also excluded from this variant as that contributed to the social correlation information included in the encoded tensor. The output from the *IE* layer is directly fed into the *PE* and then the *TMHA* layer with the dimension $D_{model}$ after the input embedding (*IE*).**MALSwoLE:** This variant of the model excludes the LSTM encoder. The hidden tensors for the LSTM decoder are prepared via some specific operations including an additional MLP that maps the $D_{model}$ out of the *TMHA* layer to a size of $D_{hid}$.

The results in Table 1 compare the above three models with our proposed architecture MALS. The improvement score is based on the mean of the difference in RMSE over the 5 s of the prediction horizon. First, it can be observed that adding the *SMHA* layer improved the model performance by 39.6%. This shows that the significance of the *SMHA* layer in extracting the social context of the traffic scenario via the graph is crucial to the prediction and is thus a vital addition to our model. Additionally, the *TMHA* layer stands as even more important, with an improvement of almost 50% by adding the layer. This confirms that, in addition to the social context, more importantly, the prediction is mainly based on the historical trajectories and self-attention to the historical trajectories which our proposed model is proficient in extracting via the *TMHA* layer. Encoding the memory information also serves as important according to our ablative results. The LSTM encoder, encoding the hidden memory information improves the model performance by 23.5%. We can also infer that part of this improvement also comes from improving the accumulative errors caused by the transformer-based *STMHA* encoder if no LSTM encoder is used to subsequently extract the hidden memory information. Figure 5 illustrates the RMSE values of the three experiments and their corresponding improvement.

Secondly, to assess the effectiveness of our proposed decoder architecture, we conduct two distinct experiments as follows.

**MALSwoLED:** This variant of the model is a version of the original transformer architecture. It excludes both the LSTM encoder and decoder. The encoded input from the *STMHA* is directly fed into another *STMHA*-based decoder similar to the transformer decoder and the right-shifted outputs are then fed into the decoder to make successive predictions.**MALSwoLDA:** This variant excludes the *STMHA* block between successive decoding steps. The hidden information from the LSTM is passed onto the next decoding step without extracting further socio-temporal context from it in making successive timestep predictions.

At first, we can observe the effects of the LSTM encoder-decoder architecture. Compared to an ordinary transformer-based encoder-decoder architecture without any RNN, we can see the errors grow almost exponentially in successive decoding steps. This is the seeming effect of the accumulative errors that originates from the autoregressive nature of the transformer decoding. In Table 2, it is clear that adding the LSTM encoding-decoding to allow hidden information passing to make predictions is superior to the transformer-based encoding-decoding, especially in future timesteps. Adding an LSTM-based encoder-decoder thus improves both the model RMSE as well as autoregressive accumulating errors. Secondly, it is also evident how the *STMHA* layer for the LSTM decoder is also crucial in mitigating successive decoding errors. Adding this layer also showcases the increasing improvement of RMSE in successive decoding steps. This establishes that extracting the socio-temporal interdependence of the traffic in successive decoding steps is very important in further predicting trajectories further into the future. Figure 6 illustrates the RMSE values of these two experiments and their corresponding improvement.

Our model also proposed a threshold to distinguish nearby vehicles from observed vehicles, which we call $d_{near}$. This threshold also limits the social influence range between vehicles, so properly designing this parameter is crucial to our proposed model. We thus also conducted further experiments to design the value of this threshold. We chose the value from a pool of four values 10,20,30,40. As shown in Table 3, excessively small values of $d_{near}$ result in poor performance seemingly due to the fact that it ignores the realistic interaction between vehicles beyond the threshold. It is also observed that values larger than 30 do not produce a notable improvement, so we chose the value of 30 to represent $d_{near}$.

### 4.6. Comparative Analysis

We use the following models in the literature to compare our model.

**CV** [16]: This method assumes a constant velocity and applies a Kalman Filter to predict future trajectories.**V-LSTM** [3]: This method uses a simple LSTM-based encoder-decoder model to make predictions.**S-LSTM** [3]: This method uses a social pooling technique to sum the neighboring vehicle features via an LSTM to predict trajectories.**CS-LSTM** [3]: This method models the traffic in grids and utilizes the convolution operation to extract social interaction and predict future trajectories.**DSCAN** [46]: This method uses a constraint network and models attention between vehicles to extract the weights to make future predictions.**SGAN** [36]: This method uses an adversarial network architecture that utilizes an encoder-decoder structure as well as a discriminator to make trajectory predictions.**HMNet** [47]: This model utilizes a hierarchical context-free LSTM encoder-decoder to forecast the trajectories.

We demonstrate the performance of our model compared to the above models in Table 4. The capability of our model to use the strength of the transformer network and model socio-temporal interaction without dealing with autoregressive errors is demonstrated in the table. Our model seemingly outperforms all other baselines, with specifically significant performance in the fourth and fifth-second prediction horizons, establishing the strength of our model in long-term predictions.

### 4.7. Prediction Visualization

RMSE is generally an effective indicator of performance but visualization is often needed to analyze the strengths and weaknesses of a model. Thus, in this section, we provide a visualization of some test cases to better understand our model’s performance in depth, by comparing predicted trajectories with ground truth. Figure 7 demonstrates three distinct scenes each with different levels of our model performance, with the trajectories marked in different colors. The first scene demonstrates a typical congested highway driving scenario. The ego here is surrounded by five other vehicles, all contributing to the interaction-modeling of the ego. With enough social interaction context, our model seems to perform very well with minimal difference in the ground truth and predicted trajectories.

The second scene represents a fairly uncongested scene compared to the first. Our proposed model also seems to perform relatively well in this case with a negligible deviation of the predicted trajectory from the ground truth. However, the performance, in contrast to the first scene, is relatively lower. We suppose it is due to the relatively much lower interaction context. Because there are not enough cars, our proposed socio-temporal interaction modeling does not produce perfectly predicted trajectories with a negligible yet noticeable deviation between the predicted and the ground truth.

The third case illustrates a scenario where there are multiple lane change cases. The performance of the model, in this case, is relatively poor due to the fact that the exact trajectory is ambiguous even though the model predicts a possible lane change. It also failed to capture the deviation of the path of the yellow vehicle due to the blue one taking its lane in front. We also believe there are not enough lane change cases in the highway dataset of BLVD which possibly also contributes to the poorer performance. We think creating a behavior prediction branch in the model to predict behaviors and then feeding the behaviors back into the model to improve the interaction prediction can improve the model’s performance on lane change scenarios, due to the fact that predicting the exact time when an intention-change will occur can improve the lane change path prediction. We also think that pre-training the model on some datasets such as HighD [48] with more lane change scenarios may mitigate some issues of extreme cases.

## 5. Conclusions

In this article, we proposed a transformer-based LSTM encoder-decoder network to model the socio-temporal interaction and predict the future trajectories of surrounding vehicles. We used the multi-head mechanism of transformers to efficiently extract the social and temporal interaction individually and have encoded it into the input as hidden information via the LSTM encoder in the encoder module. We have then used the decoder module to decode the hidden information, using another multi-head attention layer on the decoder to improve the successive decoding accuracy. We trained and evaluated our model on a practical, ego-centered, large-scale dataset that is derived from onboard sensor data. We conducted extensive studies, including both ablative and comparative studies. Our experiments verified that our model outperforms all other previous models. Ablative studies confirmed that our model solves the accumulative error caused by the transformer’s autoregressive decoding behavior. The visualization results showed our model’s strength in difficult congested scenarios, as well as its limitation in lane-change path predictions. The proposed model can be implemented in a practical driving scenario to predict the future trajectories of surrounding vehicles based on their historical tracks, provided that there is an object detection system such as YOLO is in place. In the future, we plan to enhance the model’s performance on exact lane-change tracks by better modeling the driver-intention and feeding it into the interaction procedure. As another potential future path, this model can also be extended to involve more traffic participants such as cyclists and pedestrians and predict their behavior as well. Currently, this model can only be utilized in a freeway driving scenario. Adopting this model for urban driving, incorporating more complex information such as lane types, spatial HD maps, and traffic lights may be another future direction worth pursuing.

## Figures and Tables

**Figure 1 sensors-23-00530-f001:**
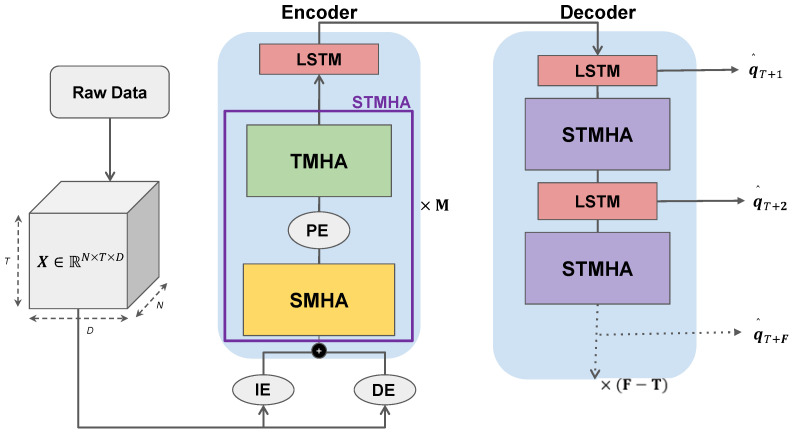
Proposed MALS-Net Architecture.

**Figure 2 sensors-23-00530-f002:**
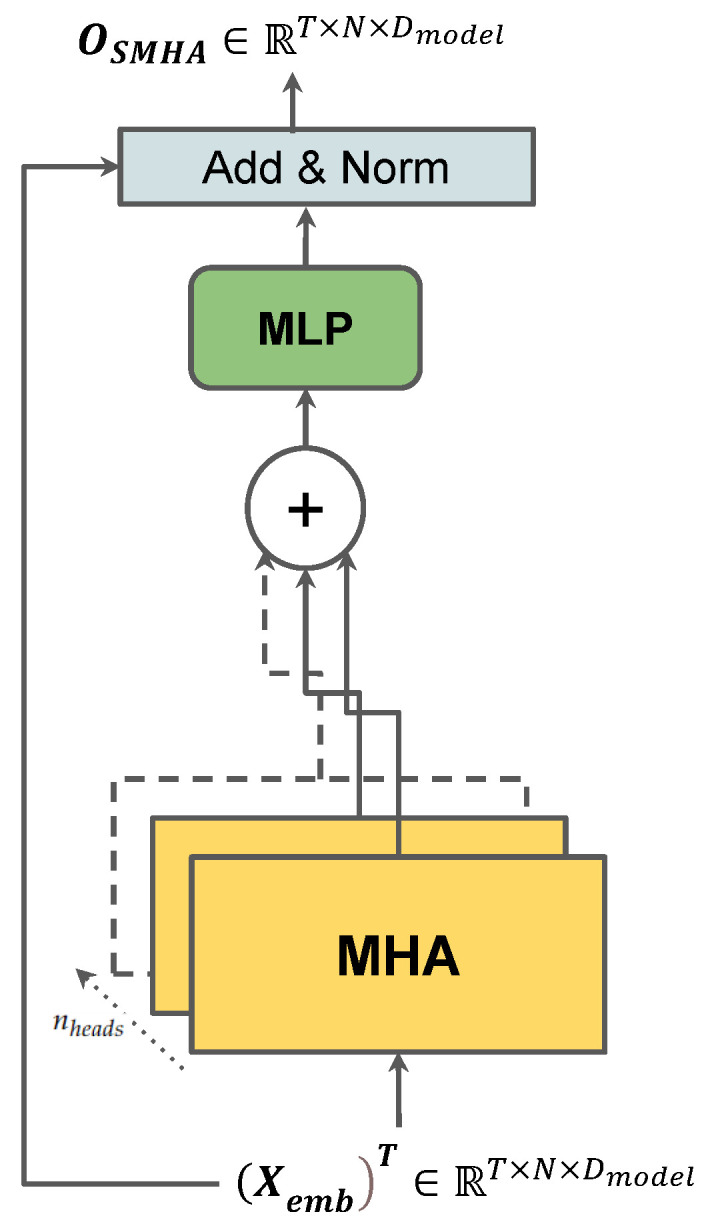
The *SMHA* block.

**Figure 3 sensors-23-00530-f003:**
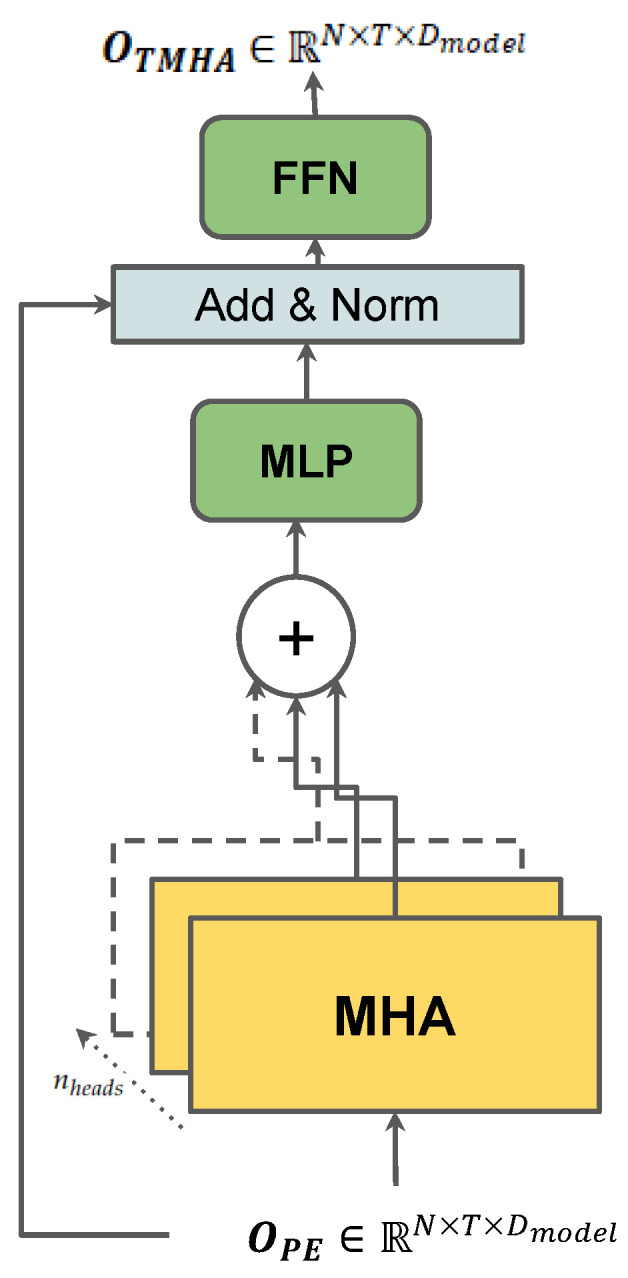
The *TMHA* block.

**Figure 4 sensors-23-00530-f004:**
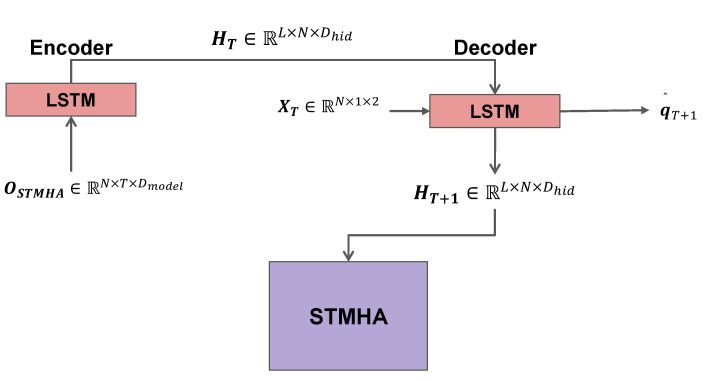
The RNN encoder-decoder block.

**Figure 5 sensors-23-00530-f005:**
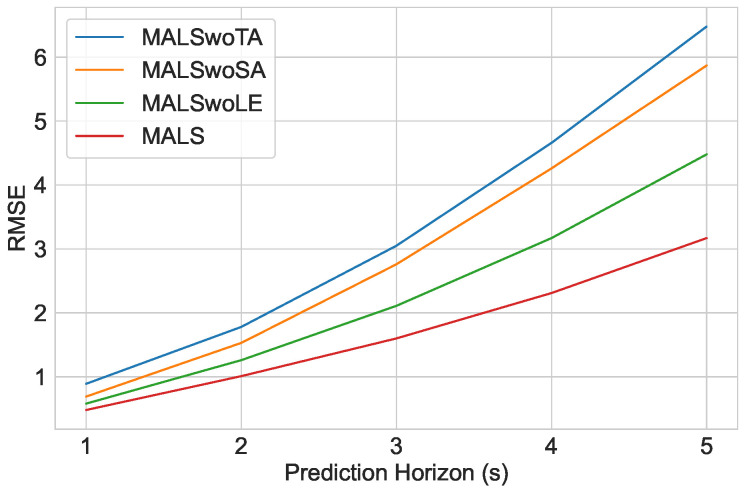
Comparison of RMSE values from the Ablative Analysis on the Encoder.

**Figure 6 sensors-23-00530-f006:**
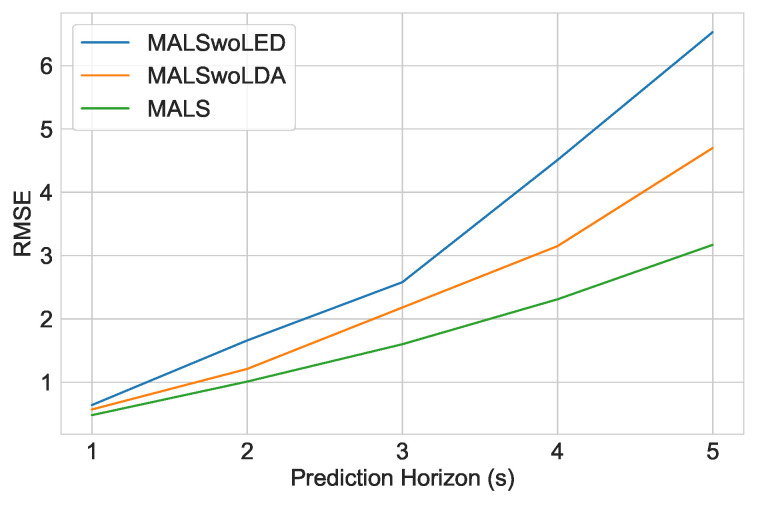
Comparison of RMSE values from the Ablative Analysis on the Decoder.

**Figure 7 sensors-23-00530-f007:**
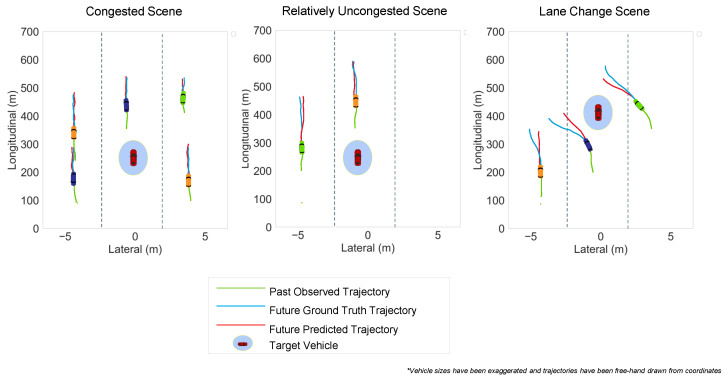
Prediction Visualization of three different scenes.

**Table 1 sensors-23-00530-t001:** Ablative Analysis of the Encoder Module.

Model	RMSE-1s	RMSE-2s	RMSE-3s	RMSE-4s	RMSE-5s	Average Improvement
MALSwoTA	0.89	1.78	3.05	4.66	6.48	47.7%
MALSwoSA	0.69	1.53	2.76	4.26	5.87	39.6%
MALSwoLE	0.58	1.26	2.11	3.17	4.48	23.5%
MALS-Net	0.48	1.01	1.60	2.31	3.36	

**Table 2 sensors-23-00530-t002:** Ablative Analysis of the Decoder Module.

Model	RMSE-1s	RMSE-2s	RMSE-3s	RMSE-4s	RMSE-5s	Average Improvement
MALSwoLED	0.64	1.66	2.58	4.71	5.53	39.1%
MALSwoLDA	0.57	1.21	2.58	3.15	4.70	25.9%
MALS-Net	0.48	1.01	1.60	2.31	3.36	

**Table 3 sensors-23-00530-t003:** Ablative Analysis of the Nearby Distance Threshold.

$d_{\mathrm{near}}$ (m) Mean	RMSE-1s	RMSE-2s	RMSE-3s	RMSE-4s	RMSE-5s
10	0.51	1.42	2.38	3.67	4.70
20	0.46	1.25	1.98	2.86	3.91
30	0.43	1.21	1.96	2.86	3.88
40	0.44	1.20	1.95	2.86	3.86

**Table 4 sensors-23-00530-t004:** Comparative Analysis.

Model Average Improvement	RMSE-1s	RMSE-2s	RMSE-3s	RMSE-4s	RMSE-5s
CV	0.73	1.78	3.13	4.78	6.68
V-LSTM	0.68	1.65	2.91	4.46	6.27
S-LSTM	0.65	1.31	2.16	3.25	4.55
CS-LSTM	0.61	1.27	2.09	3.10	4.37
DSCAN	0.58	1.26	2.03	2.98	4.13
SGAN	0.57	1.32	2.22	3.26	4.40
HMNet	0.50	1.13	1.89	2.85	4.04
MALS-Net	0.48	1.01	1.60	2.31	3.36

## Data Availability

This study was evaluated on the publicly available dataset BLVD https://drive.google.com/file/d/1fbZsIHM8Yq8TE2avDzT8xuWIBfrcFgQC/view?usp=sharing, accessed on 15 August 2022.

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
