# Peer review of "MALS-Net: A Multi-Head Attention-Based LSTM Sequence-to-Sequence Network for Socio-Temporal Interaction Modelling and Trajectory Prediction"

_sensors, 2023, doi:10.3390/s23010530_

Round 1

Reviewer 1 Report

This paper was well written and organized. The topic is quite interesting for predicting future trajectory of neighboring vehicles. The contribution is thorough. My sole concern is that the literature review is usually included in Introduction section rather than as the independent section 2 here. This is not a critical problem.

Author Response

Thank you for your comments. As for the concern about the distinction between literature review and introduction, the literature seems to be divided in this regard, with many papers in this problem choosing to use introduction to solidify the contribution of the paper while literature review is used to explain the related work and their limitations in details. Hope it is acceptable. 

Reviewer 2 Report

This paper presents a multi-head attention-based LSTM Sequence-to-Sequence model for trajectory prediction. There are two main questions in the experiment settings and results analysis section. 

Question 1: The proposed model was evaluated in the BLVD dataset. Authors should explain the data accuracy of this dataset, because the original dataset NGSIM was mentioned several times in the paper as having insufficient precision.

Question 2: The results analysis section. The authors compared their model with multiple methods, and should also compare the predicted trajectory of their model with the real trajectory.

Author Response

Thank you very much for your questions. Please find a response in the following:

  1. The NGSIM dataset was stated to be noisy and prone to overfitting issues in the paper. The accuracy was not entirely the concern. Because the two datasets have been collected by different sources of sensors and went through different levels of post-processing, the accuracy cannot be simply compared to justify the preference of one over the other. It has been explained that NGSIM was collected from a BEV camera and went through low pass filters, interpolations, etc. in order to make it usable, which introduced unrealistic trajectories in the dataset. If we train our model on that dataset, our model may get overfitted to those unrealistic instances which cause prospects of wrong predictions in a real driving scenario. However, BLVD was collected via onboard sensors from a vehicle such as lidar and camera which also has noise but the noise is much more realistic noise of the sensors and applicable to a real driving scenario. Minor revisions will however be made in the "Dataset" section to explain this slightly. Hope that will suffice. 
  2. Visualization of the predicted trajectory vs ground truth and further discussions have been provided in the "Prediction Visualization" section. Hope that suffices that concern. 

Reviewer 3 Report

Refer to the attached file, please.

Author Response

Thank you very much for your elaborate comments and concerns. Please find the response in the following.

  1. The novelty of the model has been summarized in the Introduction section. The main contributions have been solidly stated at the end of the section as an enumerated list.
  2. RMSE has been adopted by nearly all work in this area in the literature, specifically due to the fact that supervised, regression-based deep learning models are usually evaluated and validated based on RMSE, unlike probabilistic techniques which are more flexible. This is mainly a limitation of the dataset itself since most datasets split the data into training, validation, and testing in order to be evaluated via RMSE. This is done presumably so that the work is reproducible and can be easily cross-checked by peers using the same dataset. To compare and contrast the techniques with other models in the literature as well as to conduct an ablative study on the model itself, RMSE needed to be used. Hope that answers the concern, but it is always welcome to check similar work on deep learning models for trajectory prediction based on the references, in order to gain a better in-depth understanding of why RMSE was used to evaluate the model.
  3. Refer to (2)
  4. Refer to (2)
  5. Refer to (2)

Round 2

Reviewer 3 Report

I think the manuscript is suitable for publication, even though there are some shortcomings related to real-life applications.